# Voltage control of multiferroic magnon torque for reconfigurable logic-in-memory

Yahong Chai[1,10], Yuhan Liang[1,2,10], Cancheng Xiao[1,10], Yue Wang[2], Bo Li[3], Dingsong Jiang[1], Pratap Pal[4], Yongjian Tang[5], Hetian Chen[2], Yuejie Zhang[1], Hao Bai[6], Teng Xu[6], Wanjun Jiang[6], Witold Skowroński[7], Qinghua Zhang[8], Lin Gu[2], Jing Ma[2], Pu Yu[6], Jianshi Tang[1], Yuan-Hua Lin[2]✉, Di Yi[2]✉, Daniel C. Ralph[5,9], Chang-Beom Eom[4], Huaqiang Wu[1] & Tianxiang Nan[1]✉

Magnons, bosonic quasiparticles carrying angular momentum, can flow through insulators for information transmission with minimal power dissipation. However, it remains challenging to develop a magnon-based logic due to the lack of efficient electrical manipulation of magnon transport. Here we show the electric excitation and control of multiferroic magnon modes in a spin-source/multiferroic/ferromagnet structure. We demonstrate that the ferroelectric polarization can electrically modulate the magnon-mediated spin-orbit torque by controlling the non-collinear antiferromagnetic structure in multiferroic bismuth ferrite thin films with coupled antiferromagnetic and ferroelectric orders. In this multiferroic magnon torque device, magnon information is encoded to ferromagnetic bits by the magnon-mediated spin torque. By manipulating the two coupled non-volatile state variables—ferroelectric polarization and magnetization—we further present reconfigurable logic operations in a single device. Our findings highlight the potential of multiferroics for controlling magnon information transport and offer a pathway towards room-temperature voltage-controlled, low-power, scalable magnonics for in-memory computing.

In-memory computing, utilizing non-volatile memories capable of performing both information storage and logic operations within the same device, holds the promise for empowering artificial intelligence with significantly reduced energy consumption[1–4]. Existing logic-in-memory devices that have been implemented operate mainly based on charge transport, a process that inevitably gives rise to joule heating. On the other hand, information processing and transmission using magnons[5,6] as information carriers is a promising route for developing spin-based logic and memory devices[6–12] with low dissipation since magnons can transport spin in ferrimagnetic and antiferromagnetic insulators without involving moving electrons[13–17]. Incoherent magnons can be electrically (and thermally) excited in DC electronic circuits[18–21], making them compatible with current semiconductor technology. For practical applications, the implementation of magnon logic operations using gate voltages is necessary[22–25]. Current technology to manipulate magnon current transport at room temperature

[1]School of Integrated Circuits and Beijing National Research Center for Information Science and Technology (BNRist), Tsinghua University, Beijing, China. [2]School of Materials Science and Engineering, Tsinghua University, Beijing, China. [3]Institute for Advanced Study, Tsinghua University, Beijing, China. [4]Department of Materials Science and Engineering, University of Wisconsin-Madison, Madison, WI, USA. [5]Laboratory of Atomic and Solid State Physics, Cornell University, Ithaca, NY, USA. [6]Department of Physics, Tsinghua University, Beijing, China. [7]Institute of Electronics, AGH University of Science and Technology, Kraków, Poland. [8]Beijing National Laboratory for Condensed Matter Physics, Institute of Physics, Chinese Academy of Sciences, Beijing, China. [9]Kavli Institute at Cornell for Nanoscale Science, Ithaca, NY, USA. [10]These authors contributed equally: Yahong Chai, Yuhan Liang, Cancheng Xiao. ✉e-mail: linyh@tsinghua.edu.cn; diyi@mail.tsinghua.edu.cn; nantianxiang@mail.tsinghua.edu.cn

mainly relies on magnetic fields that can reorientate the magnetic ordering or modulate the magnetic domain structure[14,26].

An alternative approach involves the utilization of multiferroic materials[27–29] for magnon transport, where the magnetoelectric coupling enables the control of the magnetic order through ferroelectric switching. In the model system of multiferroic BiFeO$_3$[30,31], theoretical predictions suggest the potential for controlling the magnon dispersion by magnetoelectric coupling between the ferroelectric polarization $\overrightarrow{P}$ and Néel order $\overrightarrow{L}$[32], while experimental results demonstrated electrically tunable spin wave group velocities[33]. Recent studies also show thermally excited magnon currents in BiFeO$_3$ thin films in a longitudinal configuration that is modulated by switching the canted magnetic moment via the ferroelectric polarization[34]. These findings highlight multiferroic materials as an ideal platform for voltage-controlled magnon logic operations. However, integrable magnon-based logic devices with electrically excited incoherent magnons are yet to be realized.

Here, we propose and demonstrate a prototype multiferroic magnon-mediated spin torque (MMST) device for magnon-based reconfigurable logic operations. The device comprises multiple ferromagnetic/multiferroic BiFeO$_3$ memory cells that are positioned on a shared spin-current channel, as shown in Fig. 1a. A charge current pulse ($I_w$) flowing through the channel induces spin accumulation with polarization $\overrightarrow{\sigma}$ at the multiferroic bottom interface through the spin-Hall effect or the Rashba–Edelstein effect[35,36], which can excite antiferromagnetic magnon modes depending on the orientation of $\overrightarrow{L}$, as the spin injection transmitted to magnons is proportional to $\overrightarrow{\sigma} \cdot \overrightarrow{L}$[37,38]. As the magnons (carrying the spin-polarized angular momentum from the bottom layer) diffuse across the multiferroic layer to the ferromagnet's bottom interface, they exert the magnon torque to control the magnetic moment[17,39,40], enabling non-volatile writing of spin information to multiple cells on the current channel in parallel. For magnon logic operations, a gate voltage ($V_G$) pulse is applied across the multiferroic BiFeO$_3$ to switch $\overrightarrow{P}$, which can modulate the antiferromagnetic structure as schematically shown in the pseudo-cubic unit cell of BiFeO$_3$ (Fig. 1b). Bulk BiFeO$_3$ exhibits non-collinear antiferromagnetic order with cycloid structure due to the Dzyaloshinskii–Moriya interaction[33]. For BiFeO$_3$ thin films grown on substrates such as DyScO$_3$, the cycloid propagation direction $\overrightarrow{k}$ and $\overrightarrow{P}$ are coupled[41]. As the excited magnons in BiFeO$_3$ are proportional to $\overrightarrow{\sigma} \cdot \overrightarrow{L}$ integrated over the cycloid structure (see Supplementary Note 1), the modulation of spin cycloid structure in BiFeO$_3$ leads to a non-volatile control of the magnon spin transport.

## Results

The device heterostructure is composed of a ferromagnetic multilayer [Pt/Co]$_N$, a multiferroic BiFeO$_3$ layer, and a spin-current source SrRuO$_3$ layer which has a large spin Hall conductivity[42,43], as shown in Fig. 1c. We grew epitaxial SrRuO$_3$/BiFeO$_3$ heterostructures on orthorhombic (o) (110)$_o$ DyScO$_3$ substrates (see "Methods" for details). Atomic force microscopy reveals an atomically flat surface with a step-and-terrace structure (see Supplementary Fig. 2), and piezoelectric force microscopy (PFM) reveals a two-variant stripe ferroelectric domain structure with 71° domain walls (see Supplementary Note 2 and Supplementary Fig. 2), consistent with previous reports[41,44]. Subsequently, we deposited ferromagnetic multilayer Pt(2)/[Co(0.4)/Pt(0.92)] × 3/Co(0.4)/Pt(2) with a robust perpendicular magnetic anisotropy (numbers in parentheses indicate film thickness in nanometer, abbreviated as PtCo) onto BiFeO$_3$ (see "Methods" for details). The saturation magnetization of Co is measured to be about 1100 emu/cm$^3$ (see Supplementary Fig. 3). The cross-sectional high-angle annular dark-field scanning transmission electron microscope (HAADF-STEM) images of the tri-layer (Fig. 1d) reveals a high crystalline quality and well-defined interfaces.

To verify the device concept, we first studied the magnon transport across a multiferroic layer and the magnon torque-induced magnetization switching in an 11 nm SrRuO$_3$/120 nm BiFeO$_3$/PtCo tri-layer with a Hall-bar structure. Figure 2a illustrates the experimental setup, where the $I_w$ applied along $\overrightarrow{x}$ in SrRuO$_3$ generates a spin accumulation at the interface that excites magnon modes in BiFeO$_3$. When reaching the ferromagnetic layer PtCo, these magnons exert field-like ($\overrightarrow{\tau}_{m,FL} \propto \overrightarrow{M} \times \overrightarrow{\sigma}$) and damping-like ($\overrightarrow{\tau}_{m,DL} \propto \overrightarrow{M} \times \overrightarrow{\sigma} \times \overrightarrow{M}$) magnon torques on the magnetization $\overrightarrow{M}$, where $\overrightarrow{\sigma}$ is the spin polarization along $\overrightarrow{y}$[17]. Figure 2b shows the optical micrograph of the device and the measurement configuration. The measured anomalous Hall resistance loop (as a function of external magnetic field along $\overrightarrow{z}$, H$_z$) is depicted in Fig. 2c, confirming a perpendicular magnetic anisotropy of PtCo. Figure 2d shows the current-pulse-induced switching of perpendicular magnetization of PtCo with an in-plane magnetic field $\mu_0 H_x = 10$ mT for a deterministic magnetization switching[45]. Reversing H$_x$ results in the reversal of the magnon-torque-induced switching polarity, consistent with the symmetry of the damping-like spin-torque[46,47], ruling out possibilities of magnetization switching induced by Joule heating or Oersted field. We further excluded the self-switching of magnetization in PtCo due to the compositional gradient[48], as no current-induced switching is observed in control samples of PtCo on Si and BiFeO$_3$/PtCo bi-layer on DyScO$_3$ (see Supplementary Note 3 and Supplementary Fig. 4). Over 90% magnetization could be switched by the magnon torque, demonstrating an efficient magnon transport through multiferroic BiFeO$_3$. The threshold switching current I$_c$, defined as the current at which the normalized anomalous Hall resistance reaches 0, is determined to be 16.35 ± 0.24 mA. The error bar of the switching current is determined by the standard deviation of successive measurements (see Supplementary Note 4 and Supplementary Fig. 5). We estimate a switching current density in SrRuO$_3$ about $(3.05 ± 0.04) × 10^6$ A/cm$^2$ by a parallel resistance model (see Supplementary Note 5), which is comparable to that of the heavy metal/ferromagnetic metal system[49] and SrRuO$_3$-based heterostructures[50,51]. The linear dependence[52] of the threshold switching current on H$_x$ plotted in Fig. 2e (see Supplementary Fig. 4 for the switching hysteresis with different H$_x$) confirms again that the observed magnetization switching is governed by the magnon torque.

The magnon torque efficiency $\xi_{m,DL}$ was measured using other independent spin-torque measurements on various samples with different ferromagnetic overlayers. Through spin-torque ferromagnetic resonance (ST-FMR)[45] and second harmonic Hall voltage (SHHV) measurements[53], we estimated the $\xi_{m,DL}$ ranging from 0.012 to 0.027 in an 11 nm SrRuO$_3$/120 nm BiFeO$_3$/5 nm NiFe sample (see Supplementary Notes 6 and 7, and Supplementary Figs. 6 and 7), and 0.041 in an 11 nm SrRuO$_3$/120 nm BiFeO$_3$/PtCo sample (see Supplementary Note 8 and Supplementary Fig. 8). This $\xi_{m,DL}$ is comparable to the spin-torque efficiency in our SrRuO$_3$ and those reported by others[42,50,54], which demonstrates an efficient magnon torque generation in the tri-layers with multiferroic BiFeO$_3$ exceeding 100 nm. This is further confirmed by the BiFeO$_3$ thickness dependence of ST-FMR[55] and current-induced switching measurements (see Supplementary Note 9 and Supplementary Figs. 9 and 10).

Having established the magnon-torque-induced magnetization switching, we proceed to investigate the in-situ voltage control of magnon transport (Fig. 1a). However, in the Hall-bar devices (Fig. 2), the application of gate voltage (onto BiFeO$_3$) is not allowed, because the top (PtCo) and bottom electrodes (SrRuO$_3$) are electrically shorted through the contact electrodes. To demonstrate the in-situ voltage control of magnon torque, we etched BiFeO$_3$/PtCo layers into multiple circular micro-pillars as individual bit cells on the SrRuO$_3$ channel. The junction resistance of pillars is measured to be > 100 MΩ, suggesting the BiFeO$_3$ layer remains insulating with minimal leakage current (see Supplementary Note 10 and Supplementary Fig. 11). As illustrated in

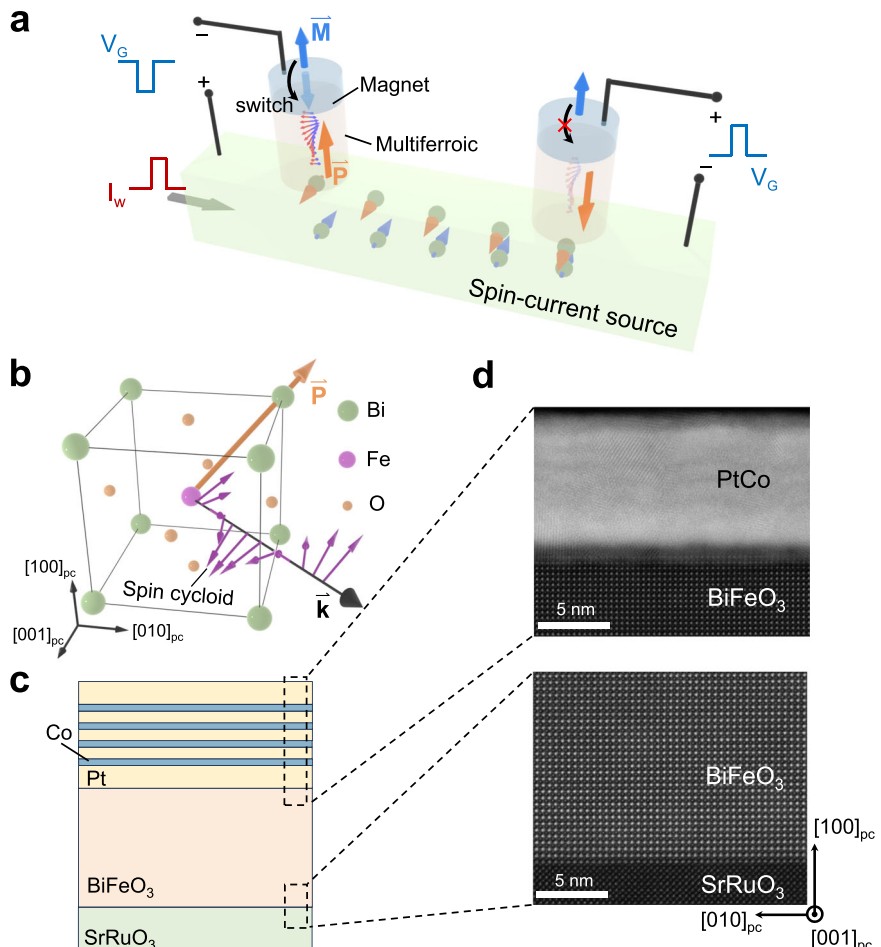

**Fig. 1 | Working principle and layered structure of the proposed multiferroic magnon spin-torque device. a** Schematic illustration of the proposed MMST device, in which ferromagnetic/multiferroic junctions are positioned on top of a spin-current source channel. An in-plane charge current pulse $I_w$ generates an out-of-plane magnon current by the spin-Hall effect that induces magnetization (blue arrow) switching through magnon torque. The magnon current can be controlled by the ferroelectric polarization (orange arrow) of the multiferroic layer by gate voltage pulses $V_G$. **b** Unit cell of $BiFeO_3$ with strongly coupled ferroelectric polarization $\vec{P}$ (orange arrow) and cycloid propagation direction $\vec{k}$ (black arrow). The magnetic moments of Fe atoms are indicated by purple arrows. **c** Structure of the $SrRuO_3/BiFeO_3/PtCo$ stack. **d** HAADF-STEM image of the $SrRuO_3/BiFeO_3/PtCo$ stack, highlighting the $PtCo/BiFeO_3$ interface (upper panel) and the $BiFeO_3/SrRuO_3$ interface (lower panel).

Fig. 3a, a global $I_w$ applied to the channel can switch the magnetization of PtCo in the cells through magnon torque, while a local voltage pulse $V_G$ applied to the selected cell across the $BiFeO_3$ layer can reverse the ferroelectric polarization. To probe the magnetization of PtCo, we employed polar magneto-optical effect (MOKE) microscopy, in which the different color contrast represents the upward/downward out-of-plane magnetization component. An optical image of the fabricated device is shown in Fig. 3b, where the probe tip on the pillar and the $SrRuO_3$ layer serve as the top and bottom electrodes for applying $V_G$. Figure 3c exhibits a well-defined hysteresis loop of the PFM signal for a $BiFeO_3/PtCo$ cell as the out-of-plane voltage is swept, demonstrating the presence of robust ferroelectricity and two distinct ferroelectric polarization states for $BiFeO_3$ in the device structure.

In Fig. 3d and e, we provide evidence of the voltage-controlled magnon torque. MOKE images were captured in the device consisting of three memory cells with different applied current pulse intervals and two distinct ferroelectric polarization states. The magnetization of the cells was initially set to $+M_z$ (bright MOKE contrast) or $-M_z$ (dark MOKE contrast) by applying out-of-plane magnetic field pulses. When the ferroelectric polarization of all three cells was initially set downwards, we found that a minimum $I_w$ of 8 mA can simultaneously switch the magnetization from $+M_z$ to $-M_z$ (or $-M_z$ to $+M_z$), depending on the polarity of $H_x$ (Fig. 3d). This behavior aligns with the magnon-torque-

induced magnetization switching discussed in Fig. 2d. To control the magnon torque in a selected cell, $V_G = -10$ V is applied to reverse the out-of-plane ferroelectric polarization component ($P_z$) of the middle cell from downwards to upwards, while the polarization of other two cells (on the sides) remain downwards. In contrast to Fig. 3d, we observed that a reduced $I_w$ of 7 mA is capable of partially switching (more than 50% of the total area) the magnetization of the middle cell, while the magnetization of the other two cells remains unswitched (Fig. 3e). This indicates a modulation of $I_c$ for magnon-torque-induced switching by the ferroelectric polarization, which is estimated to be approximately 14% (ratio = $(I_{c2}-I_{c1})/I_{c1} \times 100\%$, where $I_{c1}$ and $I_{c2}$ are the threshold currents when ferroelectric polarization is upwards or downwards, respectively). By increasing $I_w$ to 8 mA, the magnetization of all cells is switched, regardless of their ferroelectric polarization direction.

The modulation of $I_c$ is not dependent on the magnetization switching polarity, as shown in Fig. 3e (upper panel: from $+M_z$ to $-M_z$, lower panel: from $-M_z$ to $+M_z$). This observation excludes possible extrinsic effects such as magnetic domain wall pinning in the cell, which could affect the $I_c$ differently depending on the switching polarity. Additionally, we ruled out the variation of perpendicular magnetic anisotropy due to VCMA effect[56] or piezoelectric strain-mediated magnetoelectric coupling effect[57] (see Supplementary

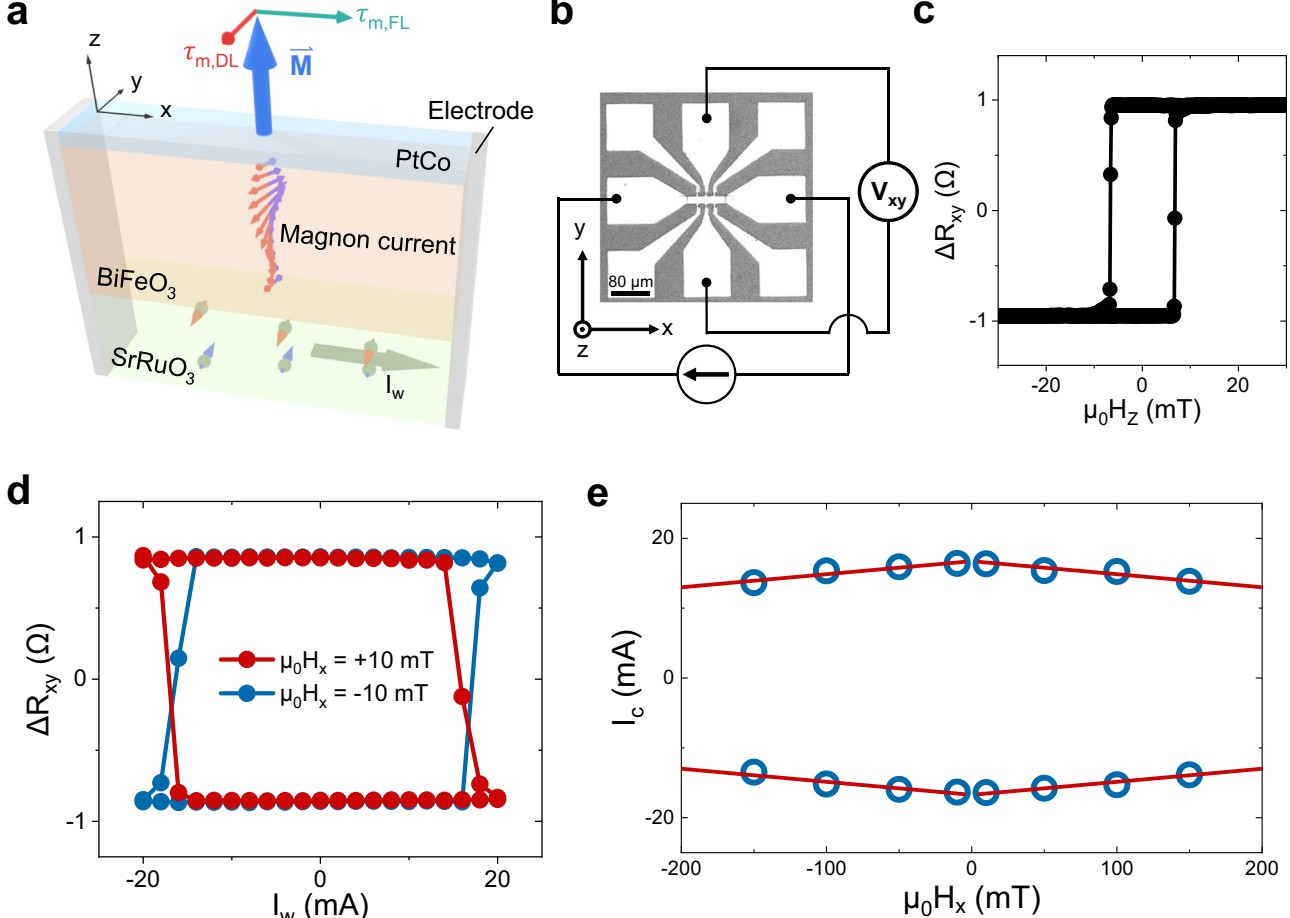

**Fig. 2 | Magnon-torque-induced switching of perpendicular magnetization at room temperature. a** Schematic diagram of the magnon torque-induced perpendicular magnetization switching. The electrodes make contact to both the $SrRuO_3$ and the PtCo layers so that the applied current flows through both layers in parallel. Red and green arrows represent the damping-like and the field-like components of magnon torque, respectively. The blue arrow denotes the magnetization of the PtCo. **b** Optical micrograph of a Hall-bar device and the measurement setup for magnon-torque-induced switching. **c** Hall resistance ($R_{xy}$) for the device of 11 nm $SrRuO_3$/120 nm $BiFeO_3$/PtCo as a function of out-of-plane magnetic field $H_z$. **d** Magnon-torque-induced switching in the device with the presence of an in-plane magnetic field $\mu_0 H_x = \pm 10$ mT. **e** Threshold current for magnetization switching $I_c$ as a function of $\mu_0 H_x$. The blue circles represent experimental data, and the red lines show the linear fitting.

Note 11), as we observed negligible variation in both coercive field and remnant magnetization on the magnetic hysteresis loop measurement of the cell before and after applying $V_G$ (see Supplementary Fig. 12). A possible mechanism of the voltage-controlled magnon torque in the $BiFeO_3$ film with two-variant domain structure could involve the change of ferroelectric domain structure driven by $V_G$[44] (see Supplementary Fig. 2 for evolution of PFM under voltages). Our model reveals that the magnon transport is quite different in the two domains separated by 71° domain walls, which have the cycloid propagation direction $\vec{k}$ orthogonal to each other (see Supplementary Note 1 and Supplementary Fig. 1).

Finally, we present the reconfigurable Boolean logic operations in a single MMST device. In the 3-terminal device configuration (Fig. 4a), the logic output ($OUT_i$) represented by the magnetization $\pm M_z$ is determined by the logic inputs of applied current $I_w$ (IN) and gate voltage $V_G$ (which controls the ferroelectric polarization acting as the synaptic weight W), as well as the initial magnetization state ($OUT_{i-1}$). By leveraging the non-volatile magnetization state ($OUT_{i-1}$) as the computational operand, a full set of 16 Boolean logic functions can be accomplished using a single MMST memory cell without necessitating changes to the circuit topology (see Supplementary Note 12 and Supplementary Fig. 13). As discussed earlier (Fig. 3), two distinct current thresholds $I_{c1}$ (=7.0 mA) and $I_{c2}$ (=8.0 mA) for switching $M_z$ can be

established depending on the ferroelectric polarization direction. As a result, an intermediate $I_w$ ($-I_{c2} < I_w < -I_{c1}$ or $I_{c1} < I_w < I_{c2}$) switches the output magnetization state only when the ferroelectric polarization is upward (W = 1). A small $I_w$ ($-I_{c1} < I_w < I_{c1}$) maintains the initial magnetization state, resulting in $OUT_i = OUT_{i-1}$, in which the initial magnetization can be set irrespective of the polarization by a large $I_w$ ($I_w > I_{c2}$ or $I_w < -I_{c2}$). Consequently, complete logic functions can be implemented and reconfigured by defining IN and supplementary steps to set the initial state $OUT_{i-1}$. As examples, we present the truth table showing settings of IN and $OUT_{i-1}$ for MMST to function as AND and XNOR logic gates in Fig. 4b, which are the common logic functions required for constructing convolutional neural network[8]. Building upon this configuration, we experimentally demonstrate the operations of a reconfigurable AND (Fig. 4c) and XNOR gate (Fig. 4d) using the $SrRuO_3$/$BiFeO_3$/PtCo device. The output magnetization is monitored using MOKE.

## Discussion

Thus far, we presented proof-of-concept experiment for reconfigurable logic computing using multiferroic magnons. We further show that the remaining reliability issue can be mitigated by using mono-domain $BiFeO_3$[56]. Supplementary Fig. 14 shows the reversible control of magnon torque by $V_G$ in a mono-domain $BiFeO_3$ sample, exhibiting a

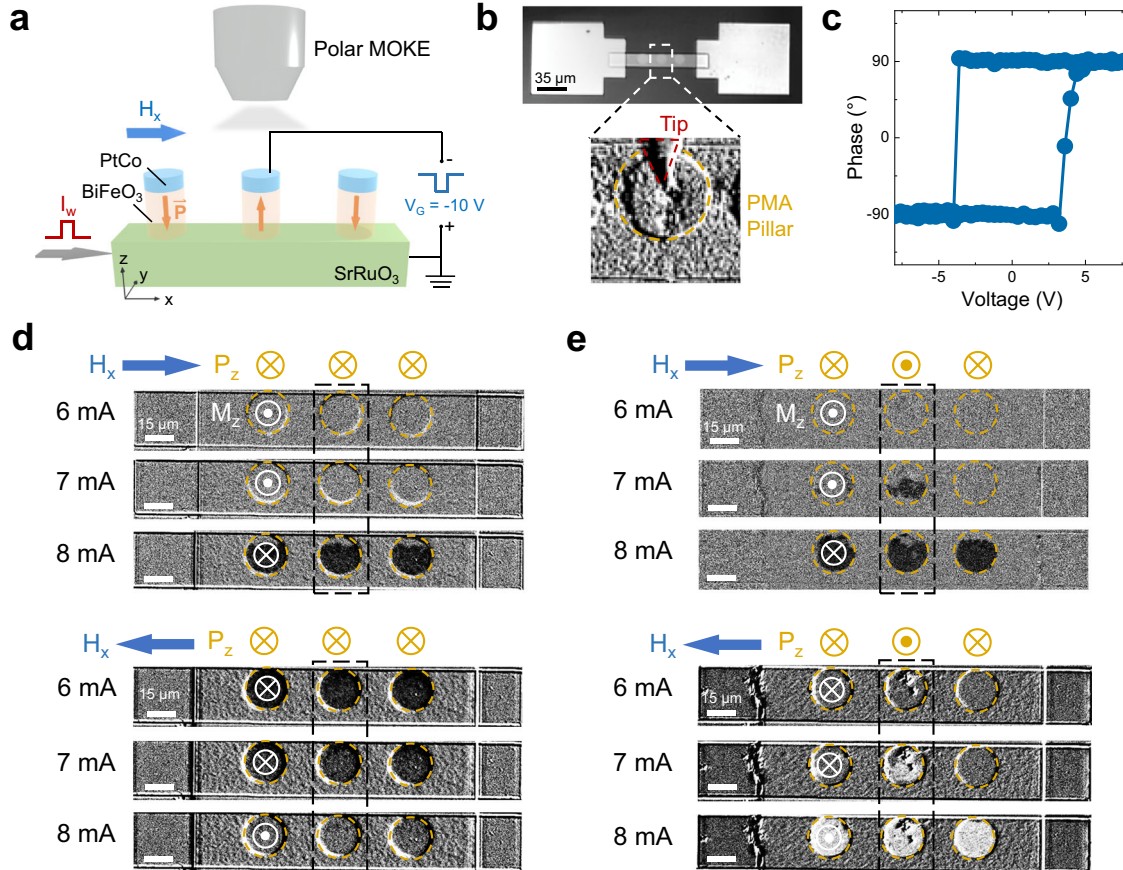

**Fig. 3 | In-situ voltage control of magnon torque. a** Schematic for the in-situ voltage control of magnon-torque-induced switching in a device comprising 120 nm thick $BiFeO_3$/PtCo memory cells on a 11 nm $SrRuO_3$ spin-current channel. Polar MOKE microscopy is used to probe the $M_z$ of the cell which gives bright ($+M_z$) and dark ($-M_z$) color contrasts. The ferroelectric polarization (orange arrow) of cells can be switched by applying $V_G$. **b** Optical micrograph of the fabricated device. The lower panel shows a zoom-in view of the cell and a probe tip employed for the application of $V_G$. **c** Out-of-plane phase signal of PFM as a function of the applied voltage. **d** and **e** MOKE images illustrating voltage-controlled magnon torque-

induced switching in three cells. The ferroelectric polarization of the middle cell circled by the black dashed box is downwards (**d**) or upwards (**e**) by applying $V_G$ before the injection of $I_w$. Yellow and white ($\odot$|$\otimes$) symbols indicate the direction of the ferroelectric polarization for $BiFeO_3$ and magnetization for PtCo, respectively. The blue arrows indicate the direction of the in-plane magnetic field $H_x$ ($\mu_0 H_x = \pm20$ mT), which determines the polarity of magnetization switching from $+M_z$ to $-M_z$ (upper panel) or from $-M_z$ to $+M_z$ (lower panel). The amplitude of $I_w$ is denoted on the left side of each image. Background subtraction during the MOKE measurement causes the different grayscale levels in different images.

modulation ratio of 4%. The smaller magnon torque modulation ratio observed in the mono-domain sample suggests other possible routes for achieving voltage-controlled magnon torque by controlling the non-collinear antiferromagnetic structures (such as the orientation of spin cycloid plane)[56]. Additionally, there could be other contributing factors in both mono-domain and two-variant-domain samples, such as ferroelectric control of spin Hall conductance in $SrRuO_3$ or the Rashba-effect at the $SrRuO_3$/$BiFeO_3$ interface, which we cannot entirely exclude. However, the different magnon torque modulation ratios in two samples indicate that the ferroelectric control of anti-ferromagnetic structure plays a major role (see Supplementary Note 13). We anticipate further improvement of the reliability and tunability of the MMST device by domain structures engineering, chemical doping, and fabrication process optimization.

For in-memory-computing applications, the utilization of magnetic tunnel junctions (MTJs) could serve as an electrical read-out mechanism for the output magnetization. The MMST-MTJs can then be incorporated in a crossbar array architecture that uses MTJ resistance summation for high throughput multiply-accumulate (MAC) operations (see Supplementary Note 14 and Supplementary Fig. 15), a fundamental process in AI. The inherent non-volatility of the ferroelectric

logic input and the magnetic logic output allows for the storage of both synaptic weights and intermediate MAC results in a non-volatile manner, distinct from other spin-based devices (see Supplementary Note 15 and Supplementary Table 1). This approach significantly minimizes memory area overhead and power consumption by reducing the necessity for intermediate calculation parameter copy and negates the requirement for data reloading after power-off. These features highlight the potential of multiferroic magnonics for low-power neuromorphic computing.

## Methods

### Sample growth and characterization

Epitaxial $SrRuO_3$/$BiFeO_3$ heterostructures were deposited on $(110)_o$-oriented $DyScO_3$ substrates using pulsed laser deposition (PLD) with a 248 nm KrF excimer laser. The $SrRuO_3$ growth was conducted at a substrate temperature of 670 °C and an oxygen partial pressure of 110 mTorr. The $BiFeO_3$ growth was conducted at a substrate temperature of 700 °C and an oxygen partial pressure of 150 mTorr. The laser fluence at target surfaces was ~1.5 J/cm² and the pulse repetition was 5–7 Hz. Subsequently, the samples were cooled to room temperature in an oxygen-rich atmosphere and transferred to a magnetron

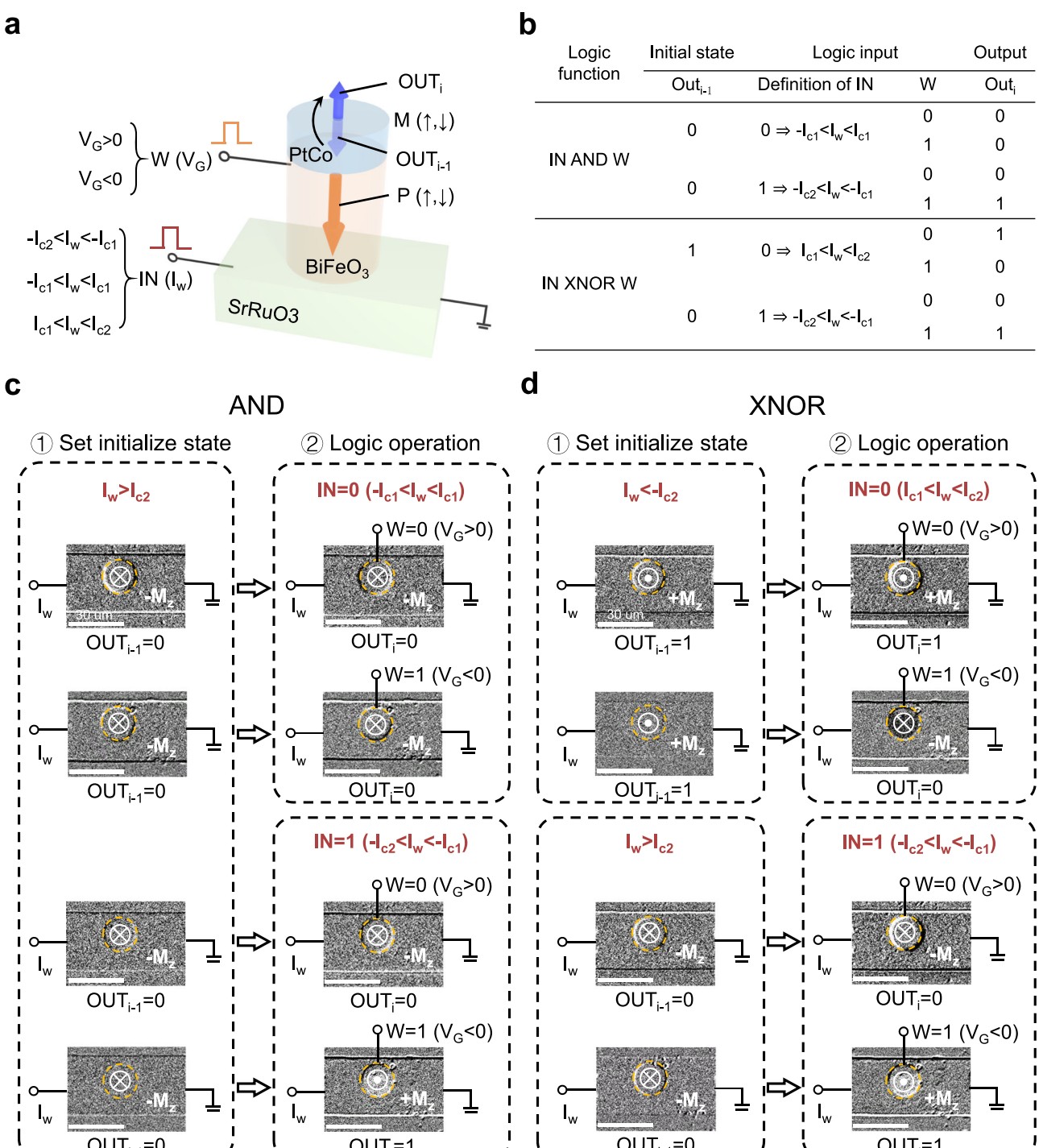

**a**

**b**

| Logic function | Initial state | Logic input | | Output |
|---|---|---|---|---|
| | $Out_{i-1}$ | Definition of IN | W | $Out_i$ |
| IN AND W | 0 | $0 \Rightarrow -I_{c1}<I_w<I_{c1}$ | 0 | 0 |
| | | | 1 | 0 |
| | 0 | $1 \Rightarrow -I_{c2}<I_w<-I_{c1}$ | 0 | 0 |
| | | | 1 | 1 |
| IN XNOR W | 1 | $0 \Rightarrow I_{c1}<I_w<I_{c2}$ | 0 | 1 |
| | | | 1 | 0 |
| | 0 | $1 \Rightarrow -I_{c2}<I_w<-I_{c1}$ | 0 | 0 |
| | | | 1 | 1 |

**Fig. 4 | Reconfigurable logic operations of the MMST device. a** Schematic illustration of the proposed MMST logic with a single memory cell positioned on a spin-current channel. The logic inputs are the applied current pulse $I_w$ (IN) and the gate voltage pulse $V_G$ (W = 0 for $V_G > 0$ and W = 1 for $V_G < 0$). The logic output is represented by the direction of $M_z$ ($OUT_i = 0$ for $-M_z$ and $OUT_i = 1$ for $+M_z$). Different logic functions can be realized by setting the initial magnetization state ($OUT_{i-1}$) and configuring different amplitudes and polarities of $I_w$ ($-I_{c2}<I_w<-I_{c1}$, $-I_{c1}<I_w<I_{c1}$ and $I_{c1}<I_w<I_{c2}$, where $I_{c1}$ and $I_{c2}$ are threshold currents for switching $M_z$ after applying $V_G < 0$ and $V_G > 0$, respectively). **b** Truth table for the reconfigurable AND and XNOR logic gates. **c** MOKE images illustrating the AND logic operations in 2 steps. Left,

MOKE images for $OUT_{i-1} = 0$, set by $I_w > I_{c2}$. Right, MOKE images demonstrate $OUT_i$ with logic inputs of "IN = 0 ($-I_{c1}<I_w<I_{c1}$), W = 0 ($V_G > 0$)", "IN = 0, W = 1 ($V_G < 0$)", "IN = 1 ($-I_{c2}<I_w<-I_{c1}$), W = 0", and "IN = 1, W = 1", respectively. **d** MOKE images illustrating the XNOR logic operations in 2 steps. Left, MOKE images for $OUT_{i-1} = 1$ set by $I_w < -I_{c2}$, and $OUT_{i-1} = 0$ set by $I_w > I_{c2}$. Right, MOKE images demonstrate $OUT_i$ with logic inputs of "IN = 0 ($I_{c1}<I_w<I_{c2}$), W = 0 ($V_G > 0$)", "IN = 0, W = 1 ($V_G < 0$)", "IN = 1 ($-I_{c2}<I_w<-I_{c1}$), W = 0", and "IN = 1, W = 1", respectively. The bright and dark contrast in the device corresponds to $+M_z$ and $-M_z$, respectively. The logic operations are performed with an in-plane magnetic field $H_x$.

sputtering chamber with a background vacuum of $1 \times 10^{-8}$ Torr for the deposition of ferromagnetic metals. The ferromagnetic multilayer PtCo or NiFe was sputter deposited at an Ar pressure of 3 mTorr. We measured the thickness of films by using X-ray reflectivity.

## Fabrication

The samples were patterned by using photolithography followed by Ar ion beam milling. Then electrodes of 100 nm Pt/5 nm Ti were deposited and defined by the lift-off process. Devices for anomalous Hall resistance (shown in Fig. 2) and SHHV (shown in Supplementary Fig. 5) measurements were patterned into 16 μm wide and 80 μm long Hall bars. Devices for ST-FMR measurements (shown in Supplementary Fig. 4) were patterned into 10 μm wide and 50 μm long microstrips with ground-signal-ground electrodes. In devices for voltage-controlled magnon torque with stripe-domain $BiFeO_3$ (shown in Figs. 3 and 4), the $SrRuO_3$ was patterned into 30 μm wide and 150 μm long microstrips and the $BiFeO_3$/PtCo were patterned into pillars with a diameter of 20–25 μm. In devices for voltage-controlled magnon torque with mono-domain $BiFeO_3$ (shown in Supplementary Fig. 8), the $SrRuO_3$ were patterned into 50 μm wide and 250 μm long microstrips and $BiFeO_3$/PtCo were patterned into pillars with a diameter of 30 μm.

## STEM characterization

For cross-sectional microscopy, sample was prepared by using focused ion beam (FIB) milling. Cross-sectional lamellas were thinned down to 60 nm thick at an accelerating voltage of 30 kV with a decreasing current from the maximum 2.5 nA, followed by fine polish at an accelerating voltage of 2 kV with a small current of 40 pA. The atomic scale HAADF-STEM images of $SrRuO_3$/$BiFeO_3$/PtCo tri-layer were performed by Cs-corrected JEM ARM200CF microscope operated at 200 kV using a high-angle annular detector for Z-contrast imaging with a collection angle of 90–370 mrad.

## Magnon-torque-induced switching measurements

In measurements of magnon torque-induced switching with Hall bars (shown in Fig. 3), pulsed currents (pulse duration of 1 ms to avoid device burnout) with various amplitudes were applied to the current channel under an external magnetic field along the current axis. After each pulse, the Hall resistance was measured with a small DC current of 0.5 mA. In measurements of magnon-torque-induced switching with the magnetization measured by a polar MOKE microscopy (shown in Fig. 4), pulsed currents (pulse duration of 1 ms) with various amplitudes were applied to the shared spin-current channel under an external magnetic field. The gate voltage pulses $V_G$ were applied to the $BiFeO_3$/PtCo cell by using the shared spin-current channel as the ground and the PtCo as the top electrode. In all measurements, the initial magnetization of PtCo was first set by an out-of-plane external magnetic field. All measurements were performed at room temperature.

## Note added to proof

Since the submission of this manuscript, we have become aware of additional results regarding ferroelectric polarization-controlled chiral spin current transport in $BiFeO_3$, which have been published in *Nat. Mater.* 23, 898-904 (2024). These findings are consistent with our observations of non-volatile voltage control of magnon torque.

## Reporting summary

Further information on research design is available in the Nature Portfolio Reporting Summary linked to this article.

## Data availability

All data needed to evaluate the conclusions in the paper are present in the paper and/or the Supplementary Materials. Additional data related to this paper may be requested from the authors. Source data are provided with this paper.

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

## Acknowledgements

This work was supported by the National Natural Science Foundation (grant numbers 52161135103, 52250418, and 62304120), the Tsinghua University Initiative Scientific Research Program, and the National Key R&D Program of China (grant number 2021YFA0716500). Worko at the University of Wisconsin–Madison was supported by the funded by Vannevar Bush Faculty Fellowship (ONR N00014-20-1-2844), the Gordon and Betty Moore Foundation's EPiQS Initiative, and grant GBMF9065 to C.B.E. Work at the Chinese Academy of Sciences was supported by the National Natural Science Foundation (grant numbers 52250402 and 52025025).

## Author contributions

T.N. and D.Y. conceived the research. T.N., H.W., C.B.E., D.C.R., D.Y., Y.H.L., J.T., W.J., P.Y., J.M., L.G. and W.S. supervised the experiments. Y.W., Y.C., P.P., Y.L., and J.M. performed the sample growth. Y.C., Y.L., D.J., Y.Z., and H.C. performed the device fabrication. Y.C., Y.L., D.J., Y.T., H.B., T.X., and H.C. performed device measurements and analysis. C.X. performed the circuit model analysis. L.G. and Q.Z. performed the high-resolution STEM experiments. B.L. performed the theoretical calculations. T.N., D.Y., Y.L., and Y.C. wrote the manuscript. All authors discussed the results and commented on the manuscript. T.N., D.Y., and Y.H.L. directed the research.

## Competing interests

The authors declare no competing interests.
