## [Peer Review File · Nature Communications]

Voltage Control of Multiferroic Magnon Torque for Reconfigurable Logic-In-MemoryREVIEWER COMMENTS

Reviewer #1 (Remarks to the Author):

In the manuscript “Voltage Control of Multiferroic Magnon Torque for Reconfigurable Logic-In-Memory”, the authors observe the magnon current excited by spin Hall effect in SrRuO₃ through multiferroic insulator BiFeO₃ can switch the perpendicular magnetized Co/Pt multilayers via magnon torque. They also show that a ferroelectric polarization switching in BFO modulates the switching threshold current and explain by a magnon diffusion change dependent on the direction of spin cycloid, which in turn is strongly coupled to the ferroelectric polarization. Finally, they propose several logic devices which use the magnetic and the electric polarization states. In general, the topic of this article is very interesting and also the experimental findings are consistent. The layout of the manuscript is also logical and clear. I would recommend the manuscript to be published in Nature Communications, if the authors can address the following questions.

1. Two series samples are utilized for voltage control of multiferroic magnon torque, one is BFO/SRO/DSO, another is BFO/SRO/STO. The modulation of switching threshold current of magnon torque is different for two series (14% and 4%, respectively). Can the authors address the reason?

2. A change of ferroelectric domain structure before and after applying voltage is observed by PFM as shown in Supplementary Fig. 2. Generally, the boundary condition is different in PtCo/BFO/SRO and AFM tip/BFO/SRO stacks and may affect the switching path of ferroelectric polarization. I suggest authors to discuss the ferroelectric polarization state in PtCo/BFO/SRO pillars.

3. Also, considering the switching of ferroelectric polarization, a strain effect could emerge and modulate the spin transmission on the BFO/SRO interface. Related magnon modes can also change due to strain effect. Can the authors discuss about or exclude this issue?

4. Although the results seem to be correlated to magnon transport in BFO layer, a contribution of electron-based spin transport should be ruled out. Since some materials are insulating in bulk, but turn semiconductor-like in the thin film limit. The authors should mention the resistivity of their BFO devices.

5. Finally, I would like to give a comment about the proposed devices. Is the multiferroic device that stores information by using ferroelectric polarization and the magnetization have any particular advantage over a system with simple two magnetic bits? I suggest the author should address this issue.

Reviewer #2 (Remarks to the Author):

In this study, Yahong Chai et al. reported voltage-controllable magnon torque in SrRuO₃/BiFeO₃/PtCoPt multilayers. They demonstrate that the spin current generated by the SrRuO₃ layer can traverse the 120 nm thick BiFeO₃ layer, subsequently switching the perpendicular magnetization of the PtCoPt multilayers. This research presents intriguing possibilities for exploring magnons in multiferroic systems. However, several critical issues need to be addressed before recommending this work.

1. Magnon Diffusion Length: The authors illustrate the magnon signal's ability to traverse the 120 nm thick BiFeO₃ layer and the 2 nm thick Pt layer to switch the magnetization of the PtCoPt multilayer. Previous studies (Ref. Science 366, 1125–1128 (2019), Phys. Rev. Appl. 11, 044070 (2019)) have shown that magnon current passing through the NiO layer is thickness-dependent, suggesting limitations based on layer thickness. It is advisable for the authors to conduct a BiFeO₃ thickness-dependent study to further support their claims.

2. PMA Layers: The authors used the PtCoPt multilayers as a PMA layer, it's worth considering the influence of the top and bottom Pt layers. Although the PtCoPt multilayer itself cannot switch PMA magnetization (confirmed in the Supplementary Information), examining the surface morphology of the 120 nm thick BiFeO₃ layer could provide insights into potential substrate differences. Additionally, exploring PMA layers without Pt, such as Ti/CoFeB/MgO, might be beneficial.

3. In this work, The damping-like torque efficiency was reported to be in the range of 0.012 to 0.027. It raises questions about achieving magnetization switching with a critical switching current density as low as 3×10^6 A/cm², particularly considering the current passing through the 120 nm thick BiFeO₃ and 2 nm thick Pt layers.

4. Does the thickness of the BiFeO₃ layer affect the critical switching current?

5. Magnetization Switching of SrRuO₃ at Low Temperatures: Given that SrRuO₃ is in a ferromagnetic state at low temperatures, is it possible to switch its magnetization at these temperatures?

6. How did the authors calculate the current density?

7. The device width is 30 μm, which seems too wide. Based on our experience, such a large width could lead to device burnout.

In conclusion, this work presents fascinating findings. I would recommend its publication if the authors address these concerns properly.

Reviewer #3 (Remarks to the Author):

See attachment

Reviewer #4 (Remarks to the Author):

In-memory computing, using non-volatile memories for storage and logic within the same device, offers energy-efficient advancements for AI. Logic-in-memory devices typically rely on charge transport, which causes joule heating, but using magnons as information carriers in spin-based devices provides a low-dissipation alternative. Magnons enable spin transport without moving electrons and can be integrated with semiconductor technology. Multiferroic materials, especially BiFeO₃ at Room temperature, allow for voltage-controlled magnon logic operations through magnetoelectric coupling, enabling control over magnon dispersion and spin wave velocities.

The authors propose a multiferroic magnon-mediated spin torque device that demonstrates reconfigurable magnon-based logic operations. The device uses spin accumulation and magnon-mediated spin torque to write spin information non-volatily across multiple cells, and it offers a promising direction for energy-efficient, high-performance computing.

The manuscript is well-written, and the results are worth publication. Before recommending it for publication in Nature Communications, this referee would like to know the authors' response to the following questions/comments.

Magnon torque-induced switching

1. Do you have only +/- 10 mT data for self-switching of magnetization in PtCo? Do you have higher field data?
2. How did you determine I_c ? What are the error bars?
3. It is not clear to the reader why SrRuO₃/BiFeO₃/NiFe is used to determine magnon torque efficiency. Can you explain?
4. Although the authors mention that efficiency is comparable to the previous studies they cited, the efficiency seems smaller than the literature. Is there any reason for the lower efficiency?

In-situ Voltage control of magnon transport

1. The value of I_c you determined in Figure 2e is larger than the minimum I_w of 8 mA in Figure 3d. Can you explain the difference?
2. Why the colors are different in Fig 3e (lower panel)?
3. Can the difference in I_c with the direction of Pz depends on the domain distribution of BiFeO₃?
4. Have you tried the same with a single-domain sample?

Reconfigurable Boolean logic operations

1. You did not define I_{c1} and I_{c2} before discussing them on page 10. What are their values for the current setup? Will those change from sample to sample? If so, which sample parameters will affect I_{c1} and I_{c2} ?
2. Does the Domain structure variation affect I_{c1} and I_{c2} ? If so, does the repetitive measurements affect I_{c1} and I_{c2} ?
3. Do the Authors have an estimate of the M_z ?

Y. Chai, et al. reported that changing the ferroelectric polarization of BFO layer in SRO/BFO/CoPt can manipulate the critical SOT switching current density of such a heterostructure. And they attribute it to the multiferroic interaction between AFM domain and ferroelectric domain in BFO, which further impact the magnon transfer process. Furthermore, they use this phenomenon to design a Boolean logic gate. Indeed, if such phenomenon related to magnon transfer dynamics can be confirmed to be true, this work can be interesting to the spintronic society. However, the author need to clarify the following concerns.

1. The changing amplitude of critical switching current (from 8 mA to 7 mA) is too small to be considered as an effective manipulation. Can the authors design experiments to enhance the manipulation ratio, e.g. changing the BFO thickness?
2. Following the first question, can this small manipulation phenomenon be caused by the ferroelectric control of magnetic anisotropy in CoPt layer? The authors didn't show the change of magnetic anisotropic energy of CoPt with different FE polarization.
3. To clarify the results, the authors should also characterize the SOT efficiency of the device with different FE polarization, which I find was missing.
4. In supplementary, two domains with polarization in in-plane and out-of-plane are assumed. It is known that the polarization direction is along [111] direction. While the BFO film in current work is [001] direction, which has in-plane component and out-of-pane component for one domain rather than two domains. Please comment this effects.
5. The control experiment where substrate/CoPt layer cannot switch itself is not convincing. I suggest to growth DSO/BFO layer under CoPt, since it might induce additional Rashba contribution. The authors need to strictly verify the SOT is indeed from SRO layer.

Overall, while the concept is new, the authors need more experiment results convince me to recommend the publication.

Response to Reviewers

We appreciate the Reviewers' interest in this work and their comments to help improve the manuscript. All reviewers acknowledged the novelty of our experimental results while posing some questions about different aspects. We have addressed all of their questions.

We have carefully considered all of the Reviewers' comments and modified the manuscript accordingly. The Reviewers' comments are reproduced below, and our responses are in the blue text. We also include a revised manuscript that contains editing markups so that our changes are easily identified.

REVIEWER 1 COMMENTS:

In the manuscript "Voltage Control of Multiferroic Magnon Torque for Reconfigurable Logic-In-Memory", the authors observe the magnon current excited by spin Hall effect in SrRuO₃ through multiferroic insulator BiFeO₃ can switch the perpendicular magnetized Co/Pt multilayers via magnon torque. They also show that a ferroelectric polarization switching in BFO modulates the switching threshold current and explain by a magnon diffusion change dependent on the direction of spin cycloid, which in turn is strongly coupled to the ferroelectric polarization. Finally, they propose several logic devices which use the magnetic and the electric polarization states. In general, the topic of this article is very interesting and also the experimental findings are consistent. The layout of the manuscript is also logical and clear. I would recommend the manuscript to be published in Nature Communications, if the authors can address the following questions.

Q1. Two series samples are utilized for voltage control of multiferroic magnon torque, one is BFO/SRO/DSO, another is BFO/SRO/STO. The modulation of switching threshold current of magnon torque is different for two series (14% and 4%, respectively). Can the authors address the reason?

Reply: The difference in the modulation of magnon current can be attributed to different modulation mechanisms in two samples which have different ferroelectric domain configurations. The DyScO₃/SrRuO₃/BiFeO₃ sample shows two-variant ferroelectric domains with each has spin cycloid with cycloid propagation directions orthogonal to each other. As we mentioned in the main text, the magnon transport through the multiferroic BiFeO₃ is quite different between the two domains (see Supplementary Fig. 1). The relatively large modulation of magnon transport in the DyScO₃/SrRuO₃/BiFeO₃ sample is mainly due to the change of ferroelectric domain structure driven by the gate voltage.

On the other hand, the SrTiO₃/SrRuO₃/BiFeO₃ sample shows mono-domain structure in which the spin cycloid propagates along only one direction. Due to the structural confinement, the application of voltage likely leads to a 180° ferroelectric switch, which does not change the cycloid propagation direction but modulates the orientation of cycloid plane (*Nat. Commun.* 8, 1583 (2017)). Therefore, the modulation of magnon transport is expected to be smaller in the SrTiO₃/SrRuO₃/BiFeO₃ sample. Additionally, there could be other contributing factors in both samples, such as ferroelectric control of spin Hall conductance in SrRuO₃ or the Rashba-effect at the SrRuO₃/BiFeO₃ interface, which we cannot entirely exclude. However, the different magnon torque modulation ratios in two samples indicate that the ferroelectric control of antiferromagnetic structure plays a major role. The issue is addressed in revised manuscript located on page 13. We summarize the differences in the two samples in the **Table 1** below.

Table 1. The differences between DyScO₃/SrRuO₃/BiFeO₃/PtCo and SrTiO₃/SrRuO₃/BiFeO₃/PtCo heterostructures

Sample	DyScO ₃ /SrRuO ₃ /BiFeO ₃ /PtCo	SrTiO ₃ /SrRuO ₃ /BiFeO ₃ /PtCo
Coercive field	20 mT	22 mT
Threshold current density	3×10 ⁶ A/cm ²	3.2×10 ⁶ A/cm ²
Modulation ratio	14%	4%
Domain type	Two-variant domain	Mono-domain
Modulation mechanism	Modulation of two-variant domain	Modulation of cycloid plane

Q2. A change of ferroelectric domain structure before and after applying voltage is observed by PFM as shown in Supplementary Fig. 2. Generally, the boundary condition is different in PtCo/BFO/SRO and AFM tip/BFO/SRO stacks and may affect the switching path of ferroelectric polarization. I suggest authors to discuss the ferroelectric polarization state in PtCo/BFO/SRO pillars.

Reply: We agree with the reviewer that the boundary condition for ferroelectric polarization switching is important and differs between SrRuO₃/BiFeO₃/PtCo pillar devices and during the PFM measurement. The observed differences can be attributed to the drag field effect or other extrinsic effects due to the scanning of PFM probe tip. Due to the screening effect of the relative thick PtCo layer and additional top capping

metal, it is extremely challenging to directly measure the ferroelectric domains in our SrRuO₃/BiFeO₃/PtCo pillars using PFM. This issue has already been addressed in a previous study by etching the top metals after the ferroelectric switching. The results show that the evolution of ferroelectric domains of BiFeO₃ with the similar two-variants domain by using PFM tip is almost the same as by using top electrode in SrRuO₃/BiFeO₃/CoFe stack (*Nature* 516, 370-373 (2014)). This previous report supports our observations.

Q3. Also, considering the switching of ferroelectric polarization, a strain effect could emerge and modulate the spin transmission on the BFO/SRO interface. Related magnon modes can also change due to strain effect. Can the authors discuss about or exclude this issue?

Reply: We agree with the reviewer that the piezoelectric strain can be induced in ferroelectric materials such as in BiFeO₃ (*Nat. Commun.* 11, 1704 (2020)). However, we believe that strain does not play a major role in our devices for the following reasons. First, the amount of strain induced in our samples should be negligible because of the substrate clamping effect. Moreover, the modulation of magnon transport behavior is observed in the remnant states after removing the voltage. Considering the domain architectures and 180° ferroelectric polarization switch in each domain, the remnant strain is expected to be almost the same. Furthermore, the change of magnetic anisotropy can serve as a sensitive probe for the existence of strain. In Supplementary Figure 10, no clear change of the magnetic anisotropy in PtCo is observed before and after applying gate voltages. This indicates a negligible strain effect, as well as a negligible voltage-controlled magnetic anisotropy (VCMA) effect. We have included this discussion in revised manuscript (see page 9) and Supplementary Note 12.

Q4. Although the results seem to be correlated to magnon transport in BFO layer, a contribution of electron-based spin transport should be ruled out. Since some materials are insulating in bulk, but turn semiconductor-like in the thin film limit. The authors should mention the resistivity of their BFO devices.

Reply: We thank the referee for this insightful comment. To rule out the contribution of spin transport mediated by electrons, we conducted leakage current measurement in BiFeO₃ thin films. In the same device shown in Fig. 3 (25- μ m-diameter 11 nm SrRuO₃/120 nm BiFeO₃/PtCo circular micro-pillar), we measured the current-voltage (I-V) characteristic curve using Keithley 4200A-SCS Parameter Analyzer. The leakage current is estimated to be about 10⁻⁸ A at 1 V (see **Figure R2**), corresponding to a junction resistance of about 10⁸ Ω and a resistivity about 4.1 \times 10⁷ Ω ·cm for the 120 nm BiFeO₃ film, which indicates our 120 nm BiFeO₃ films are good insulators.

Therefore, we rule out the contribution from the electron mediated spin transport. These results are now also mentioned in revised manuscript (see page 8) and included in Supplementary Materials (see Supplementary Figure 11).

Figure R2. The measured leakage current versus out-of-plane voltage in the device with 120 nm BiFeO₃/PtCo circular micro-pillars (25 μm diameter) stack on 11 nm SrRuO₃ channel.

Q5. Finally, I would like to give a comment about the proposed devices. Is the multiferroic device that stores information by using ferroelectric polarization and the magnetization have any particular advantage over a system with simple two magnetic bits? I suggest the author should address this issue.

Reply: Thanks for the reviewer for this insightful comment. From a memory device perspective, the integration of ferroelectric polarization and magnetization offers no distinct advantages over a system with two magnetic bits. However, from the perspective of in-memory computing, the ability to manipulate magnetization by ferroelectric polarization in a non-volatile manner (whereas VCMA is a volatile modulation) empowers a single device to perform all 16 Boolean logic operations within two steps, which cannot be achieved by a system with two magnetic states (*Microelectronics Journal* 131, 105635(2023)). This potentially broadens the computational capabilities of crossbars based on emerging non-volatile memories. Meanwhile, more complex logical functions can be further realized based on the simple Boolean logic operations, which also elevates the edge of MRAM in digital in-memory computing. We have summarized the device properties and functionalities between our proposed in-memory logic device and other spin-based devices in **Table 2**. We also envision great potential for such non-volatile control in future

neuromorphic computing. We have included this discussion in Supplementary Materials (see Supplementary Note 15).

Table 2. Comparison of different emerging nonvolatile memories with logic-in-memory architecture.

Device	Write information	Gate controllability	Logic function	Reference
Spin-transfer torque MRAM	Spin-transfer torque	N/A	N/A	Nature 601, 211–216 (2022)
Spin-orbit torque MRAM	Spin-orbit torque	N/A	N/A	Science 336, 555–558 (2012)
Voltage-controlled spin–orbit torque	Spin-orbit torque	Voltage-control of magnetic anisotropy, volatile	XNOR logic (with two devices)	Nat. Electron. 1, 398–403 (2018)
Magnetoelectric spin–orbit logic (proposed)	Magnetoelectric control of magnetic vector	N/A	Buffer logic (with single device)	Nature 565, 35–42 (2019)
Multiferroic magnon spin-torque device	Magnon-mediated spin torque	Magnetoelectric control of magnon transport, non-volatile	Reconfigurable 16 Boolean logics (with single device)	This work

REVIEWER 2 COMMENTS:

In this study, Yahong Chai et al. reported voltage-controllable magnon torque in SrRuO₃/BiFeO₃/PtCoPt multilayers. They demonstrate that the spin current generated by the SrRuO₃ layer can traverse the 120 nm thick BiFeO₃ layer, subsequently switching the perpendicular magnetization of the PtCoPt multilayers. This research presents intriguing possibilities for exploring magnons in multiferroic systems. However, several critical issues need to be addressed before recommending this work.

Q1. Magnon Diffusion Length: The authors illustrate the magnon signal's ability to traverse the 120 nm thick BiFeO₃ layer and the 2 nm thick Pt layer to switch the magnetization of the PtCoPt multilayer. Previous studies (Ref. Science 366, 1125–1128 (2019), Phys. Rev. Appl. 11, 044070 (2019)) have shown that magnon current passing through the NiO layer is thickness-dependent, suggesting limitations based on layer thickness. It is advisable for the authors to conduct a BiFeO₃ thickness-dependent study to further support their claims.

Reply: We thank the reviewer for this insightful comment. To reveal the BiFeO₃ thickness dependence of magnon transport, we have included additional ST-FMR results in 11 nm SrRuO₃/BiFeO₃/5 nm NiFe samples with various thicknesses of BiFeO₃ ranging from 20 nm to 120 nm (for $t_{\text{BFO}} = 20, 40, 60$ and 120 nm). **Figure R3** shows the damping-like magnon torque efficiency $\xi_{m,DL}$ as a function of BiFeO₃ thickness, indicating a non-monotonic trend similar to that observed in NiO-based heterostructures (*Phys. Rev. Appl.* 11, 044070 (2019)). We also include the mentioned references to support our demonstration and clarify this result in our revised manuscript (see page 7) and Supplementary Materials (see Supplementary Note 9 and Supplementary Figure 9).

Figure R3. Magnon torque efficiency $\xi_{m,DL}$ (measured by ST-FMR) of 11 nm SrRuO₃/BiFeO₃/5 nm NiFe samples with various BiFeO₃ thicknesses $t_{\text{BFO}} = 20, 40, 60$ and 120 nm.

Q2. PMA Layers: The authors used the PtCoPt multilayers as a PMA layer, it's worth considering the influence of the top and bottom Pt layers. Although the PtCoPt multilayer itself cannot switch PMA magnetization (confirmed in the Supplementary Information), examining the surface morphology of the 120 nm thick BiFeO₃ layer could provide insights into potential substrate differences. Additionally, exploring PMA layers without Pt, such as Ti/CoFeB/MgO, might be beneficial.

Reply: We thank the reviewer for this comment. We chose the PtCo films as the ferromagnetic layer because PtCo exhibits robust PMA regardless of different surfaces and interfaces. As we have shown in Supplementary Fig. S3, the multilayers on Si with symmetric top and bottom Pt layers do not show self-switching behaviors. We agree with the Reviewer that the 120 nm BiFeO₃ layer could potentially lead to a different interface/surface environment compared to Si, if the surface morphology is very rough. However, as shown in **Figure R4** below, the surface morphology of our DyScO₃/11 nm SrRuO₃/120 nm BiFeO₃ sample shows an atomically flat surface with a step-and-terrace structure and surface roughness better than 0.3 nm. Therefore, it is highly unlikely that the BiFeO₃ surface morphology would lead to self-switching torque. We have addressed the AFM result in the revised manuscript (located at page 4) and included into revised Supplementary Figure 2.

Figure R4. Atomic force microscopic image of a DyScO₃/ 11 nm SrRuO₃/ 120 nm BiFeO₃ sample. The scale bar is 500 nm.

To further rule out the self-switching torque arising due to the BiFeO₃/PtCo interface, we fabricated DyScO₃/ 120 nm BiFeO₃/PtCo sample (without SrRuO₃) and conducted current switching experiments. As shown in **Figure R5**, no switching can be observed, further validating our conclusion. We have included the results of BiFeO₃/PtCo sample in the revised Supplementary Figure 4.

Figure R5. The absence of current switching in DyScO₃/ 120 nm BiFeO₃/ PtCo control sample. **a**, Schematic of the layered structure for the PtCo on BiFeO₃ thin film without SrRuO₃. **b**, The anomalous Hall resistance ΔR_{xy} - I_w loops for PtCo on BiFeO₃.

As the Reviewer suggested, we tried to fabricate 3 nm Ti/ 1nm CoFeB/ 1.2 nm MgO multilayers which are typical structures with perpendicular magnetic anisotropy on BiFeO₃. Unfortunately, this sample does not show perpendicular magnetic anisotropy as shown by the hysteresis loop of MOKE signal in **Figure R6**. Further optimization is needed to integrate other perpendicularly magnetized ferromagnetic thin films beyond PtCo.

Figure R6. Magnetic hysteresis loop of a 3 nm Ti/ 1nm CoFeB/ 1.2 nm MgO layer stack on DyScO₃/ 11 nm SrRuO₃/ 120 nm BiFeO₃, measured by using MOKE.

Q3. In this work, the damping-like torque efficiency was reported to be in the range of 0.012 to 0.027. It raises questions about achieving magnetization switching with a critical switching current density as low as 3×10^6 A/cm², particularly considering the current passing through the 120 nm thick BiFeO₃ and 2 nm thick Pt layers.

Reply: We thank the reviewer for this comment. We acknowledge that the correlation

between the threshold switching current density and the damping-like torque efficiency might appear complex, especially in micrometer-scale devices. According to the recent studies (*Phys. Rev. Appl.* 15, 024059 (2021).), the simple correlation between the threshold switching current density and the damping-like torque efficiency might be unreliable. In previous works on SrRuO₃, the damping-like torque efficiency was reported to be about 0.03 in NiFe/SrRuO₃ heterostructures on similar orthorhombic NdGaO₃ substrates (*Adv. Mater.* 33, 2007114 (2021).). The critical switching current density was reported to be 4×10^6 A/cm² in CoPt/SrRuO₃ heterostructures (*Phys. Rev. Mater.* 7, 024418 (2023)) and 4.5×10^6 A/cm² in FeGd/SrRuO₃ heterostructures (*Adv. Electron. Mater.* 8, 2200514 (2022)). Both the damping-like magnon torque efficiency and the switching current density reported in our work are within the same order of magnitude as those in previous reports. This discussion has been included in the revised manuscript, see page 7.

Q4. Does the thickness of the BiFeO₃ layer affect the critical switching current?

Reply: We thank the reviewer for this comment. We have conducted additional magnon torque switching experiments in 11 nm SrRuO₃/ 42 nm BiFeO₃/PtCo sample. As shown in **Figure R7**, we observe the magnon-torque-induced magnetization switching with the critical switching current density of about 2.8×10^6 A/cm². This value is slightly lower than that of the 11 nm SrRuO₃/120 nm BiFeO₃/PtCo sample (reported in Fig. 2), which might be attributed to the enhanced magnon transport efficiency in samples with thinner BiFeO₃ layers. We have also included these results in revised manuscript (see page 7) and in Supplementary Materials (see Supplementary Note 9 and Supplementary Figure 10).

Figure R7. Magnon-torque-induced switching in an 11 nm SrRuO₃/42 nm BiFeO₃/PtCo tri-layer. **a**, Anomalous Hall resistance loop as sweeping out-of-plane magnetic field. **b**, Magnon-torque-induced switching with the presence of an in-plane magnetic field $\mu_0 H_x = \pm 10$ mT.

Q5. Magnetization Switching of SrRuO₃ at Low Temperatures: Given that SrRuO₃ is in a ferromagnetic state at low temperatures, is it possible to switch its magnetization at these temperatures?

Reply: We agree with the reviewer's comment. In our device, the SrRuO₃ is the spin-current source that generates the spin current at room-temperature. As mentioned by the Reviewer, SrRuO₃ becomes ferromagnetic below 150 K. Magnon torque can indeed induce the switching of SrRuO₃ magnetization using other spin-current source. This has been observed in a very recent report involving SrRuO₃/BiFeO₃/SrIrO₃ heterostructures at low temperature, with SrIrO₃ as the spin-source layer and SrRuO₃ as the ferromagnetic layer (*Nat. Mater.* (2024), <https://doi.org/10.1038/s41563-024-01854-8>).

Q6. How did the authors calculate the current density?

Reply: We thank the reviewer for this comment. In Figure 2, we show the current switching of PtCo in a Hall bar device, in which the current density is calculated according to parallel resistance model. Our control samples provide the resistivity for the calculation: the resistivity of 8.3 nm PtCo layer on Si/SiO₂ substrate ρ_{PtCo} is estimated to be about 120 $\mu\Omega\cdot\text{cm}$, and the resistivity of 11 nm SrRuO₃ layer on DyScO₃ substrate ρ_{SRO} is about 325 $\mu\Omega\cdot\text{cm}$. The current density in SrRuO₃ layer is determined by the following expression:

$$j_{SRO} = \frac{I_w}{W} \times \frac{\rho_{PtCo}}{\rho_{SRO} \cdot d_{PtCo} + \rho_{PtCo} \cdot d_{SRO}}$$

where the I_w is the magnitude of current pulse, $W = 16 \mu\text{m}$ is the width of device, $t_{PtCo} = 8.3 \text{ nm}$ is the thickness of PtCo layer, and $d_{SRO} = 11 \text{ nm}$ is the thickness of SrRuO₃ layer. In Figure 2, the threshold current $I_c = 16 \text{ mA}$, corresponding to $j_{SRO} \approx 3.0 \times 10^6 \text{ A/cm}^2$.

In Figure 3 we show the current switching of PtCo in pillar devices. The current density is calculated as $j_{SRO} = \frac{I_w}{W \cdot d_{SRO}} \approx 2.4 \times 10^6 \text{ A/cm}^2$ ($I_c = 8 \text{ mA}$, $W = 30 \mu\text{m}$) because there is no current shunting in PtCo layer in these devices. We find the critical current densities in the Hall bar (Figure 2) and pillar devices (Figure 3) are consistent. We also include these discussions and calculations in our revised manuscript (located on page 6) and Supplementary Materials (see Supplementary Note 6).

Q7. The device width is 30 μm , which seems too wide. Based on our experience, such

a large width could lead to device burnout.

Reply: We thank the reviewer for this insightful comment. The size of the device is currently limited by the fabrication process of pillar devices and the practicality of using a probe tip for in-situ gating measurements. Specifically, the probe needs to be landed on the pillar for ferroelectric switch measurements, which requires the device width to be at least 20 μm by using our instruments. To mitigate the possibility of device burnout, we have taken measures such as using current pulses with pulse width no longer than 1 ms to avoid extensive Joule heating. We also address this issue in our revised manuscript, located on page 19.

In conclusion, this work presents fascinating findings. I would recommend its publication if the authors address these concerns properly.

We have addressed the reviewer's concern accordingly, and we hope the reviewer now standing for recommending our works publication in Nature Communications.

REVIEWER 3 COMMENTS:

Y. Chai, et al. reported that changing the ferroelectric polarization of BFO layer in SRO/BFO/CoPt can manipulate the critical SOT switching current density of such a heterostructure. And they attribute it to the multiferroic interaction between AFM domain and ferroelectric domain in BFO, which further impact the magnon transfer process. Furthermore, they use this phenomenon to design a Boolean logic gate. Indeed, if such phenomenon related to magnon transfer dynamics can be confirmed to be true, this work can be interesting to the spintronic society. However, the author need to clarify the following concerns.

Q1. The changing amplitude of critical switching current (from 8 mA to 7 mA) is too small to be considered as an effective manipulation. Can the authors design experiments to enhance the manipulation ratio, e.g. changing the BFO thickness?

Reply: We thank the reviewer for this insightful comment. Firstly, we would like to highlight the advantages of our approach compared to other available methods, such as voltage-controlled magnetic anisotropy (VCMA), for achieving the voltage control of critical switching current. In the state-of-art VCMA, the typical modulation ratio of threshold current is about 20%, corresponding to a tunability of 1% per 4 mV/nm (*Appl. Phys. Lett.* 118, 052409 (2021)). In this work, we have achieved a modulation ratio for threshold current of 14% with an applied voltage of 10 V, corresponding to a tunability of 1% per 0.7 mV/nm. This demonstrates a comparable modulation ratio but with a larger (~6 times) turnability than VCMA. Additionally, the non-volatile modulation of our approach also allows us to design and operate the reconfigurable logic-in-memory devices. Therefore, we believe the ferroelectric switching control of magnon torque could be an effective approach to manipulate the critical current.

Moreover, we note that a very recent report on similar SrRuO₃/BiFeO₃/SrIrO₃ structures show a consistent modulation ratio, in which the magnon torque critical switching current can be modulated from 3 mA to 4 mA by controlling the ferroelectric polarization of BiFeO₃ (*Nat. Mater.* 2024, <https://doi.org/10.1038/s41563-024-01854-8>). This result further validates our conclusion.

Regarding the enhancement of the manipulation ratio, we acknowledge that the modulation ratio can potentially be further improved by carefully engineering the antiferromagnetic structure of BiFeO₃, because the magnon transport is strongly correlated to the nature of antiferromagnetic order as discussed in Supplementary Note 1. This includes controlling crystalline structure, orientation, and chemical composition, which is currently under investigation and beyond the scope of this manuscript.

Q2. Following the first question, can this small manipulation phenomenon be caused

by the ferroelectric control of magnetic anisotropy in CoPt layer? The authors didn't show the change of magnetic anisotropic energy of CoPt with different FE polarization.

Reply: We thank the reviewer for this comment. As shown in Supplementary Fig. 10, the perpendicular magnetic anisotropy (PMA) in PtCo layer is very robust and show negligible modulation of coercive field (H_c remains around 20 mT before and after applying voltage) and ferromagnetic remanence (M_r/M_s remains around 0.97-0.98 before and after applying voltage) in the pillar devices. We have included this discussion in revised manuscript (located on page 9) and Supplementary Materials (see Supplementary Note 11).

Q3. To clarify the results, the authors should also characterize the SOT efficiency of the device with different FE polarization, which I find was missing.

Reply: We thank the reviewer for this insightful comment. In this work, the quantification of SOT efficiency in our heterostructures is carried out by conducting standard spin-orbit torque measurements (including both spin-torque ferromagnetic resonance ST-FMR and second harmonic Hall voltage SHHV), with the experimental setups shown in Supplementary Fig. 6a and Supplementary Fig. 7a, respectively. However, both device configurations do not allow the in-situ application of bias voltage to switch the ferroelectric polarization, because the top (PtCo) and bottom electrodes (SrRuO₃) are electrically shorted through the contact electrodes.

While in our devices with BiFeO₃/PtCo pillars stack on spin-source channel, the observation of voltage-controlled switching threshold current indicates the torque efficiency to be modulated by ferroelectric polarization. However, it is hard to characterize the torque efficiency in such pillar device configuration by ST-FMR, SHHV or other standard spin-torque measurement techniques. This is because the PtCo in pillars is electrically isolated from the current source, and thus traditional methods such as ST-FMR (relies on anisotropic magnetoresistance) and SHHV (relies on Hall effect) cannot be utilized to in-situ characterize the torque efficiency. We acknowledge the importance of the direct characterization of torque efficiency in the pillar devices and are exploring alternative methods and device configurations to address this aspect in future studies. We have included these considerations and limitations in our revised manuscript for clarity and completeness, located on page 8.

Q4. In supplementary, two domains with polarization in in-plane and out-of-plane are assumed. It is known that the polarization direction is along [111] direction. While the BFO film in current work is [001] direction, which has in-plane component and out-of-plane component for one domain rather than two domains. Please comment this effects.

Reply: We appreciate the reviewer's attention to details and would like to clarify the depiction in Supplementary Fig. 1c to avoid any confusion. Supplementary Fig. 1c is a schematic top view of the BiFeO₃ films with two-variant domains (pink and yellow regions). The bold yellow arrows (labeled as P) correspond to the in-plane components of ferroelectric polarization in each domain, which is orthogonal to each other with 71° domain walls. We apologize for any misunderstanding caused and have clearly described this in the revised figure caption.

In rhombohedral BiFeO₃, the spin cycloid direction (\vec{k} in Supplementary Fig. 1c) is orthogonal to the in-plane component of ferroelectric polarization (*Nature* 549, 252-256 (2017)). Therefore, the magnon transport through BiFeO₃ is expected to be quite different between the two domains, depending on the relative angle between spin polarization of spin current and the \vec{k} . In [001] direction BiFeO₃ with two-variant domain structure, the ferroelectric polarization is along [111] with both in-plane and out-of-plane components. Therefore, the application of gate voltage also affects the in-plane component. As shown in Supplementary Fig. 1, the relative population of the two domains is modulated after applying voltage, leading to potential changes in magnon transport behaviors, as discussed in Supplementary Note 1. We have included these discussions and clarification in our revised Supplementary Materials (located at Supplementary Note 1 and Supplementary Figure 1) for better understanding.

Q5. The control experiment where substrate/CoPt layer cannot switch itself is not convincing. I suggest to growth DSO/BFO layer under CoPt, since it might induce additional Rashba contribution. The authors need to strictly verify the SOT is indeed from SRO layer.

Reply: We appreciate the reviewer's input and agree that verifying the absence of self-switching effects in the control experiment is crucial. To address this concern, we have fabricated the 120 nm BiFeO₃/PtCo bi-layer on (110)_o-oriented DyScO₃ substrate, and conducted the current switching experiments. As shown in **Figure R8** below, we did not observe magnetization switching in the control sample, even with currents up to 20 mA and $H_x=|10\text{mT}|$. These results rule out the possibility of magnetization switching due to Rashba effect at the BiFeO₃/PtCo interface and the self-torque effect in PtCo. These findings have been incorporated into our revised manuscript (see page 6) and Supplementary Materials (see Supplementary Note 3 and Supplementary Figure 4).

Figure R8. The absence of current switching in DyScO₃/120 nm BiFeO₃ /PtCo control sample. a, Schematic of the layered structure for the PtCo on BiFeO₃ thin film without SrRuO₃. b, The anomalous Hall resistance ΔR_{xy} - I_w loops for PtCo on BiFeO₃.

Overall, while the concept is new, the authors need more experiment results convince me to recommend the publication

Reply: we have addressed the reviewer's concern accordingly, and we hope the reviewer now standing for recommending our works publications in Nature Communications.

REVIEWER 4 COMMENTS:

In-memory computing, using non-volatile memories for storage and logic within the same device, offers energy-efficient advancements for AI. Logic-in-memory devices typically rely on charge transport, which causes joule heating, but using magnons as information carriers in spin-based devices provides a low-dissipation alternative. Magnons enable spin transport without moving electrons and can be integrated with semiconductor technology. Multiferroic materials, especially BiFeO₃ at Room temperature, allow for voltage-controlled magnon logic operations through magnetoelectric coupling, enabling control over magnon dispersion and spin wave velocities.

The authors propose a multiferroic magnon-mediated spin torque device that demonstrates reconfigurable magnon-based logic operations. The device uses spin accumulation and magnon-mediated spin torque to write spin information non-volatilely across multiple cells, and it offers a promising direction for energy-efficient, high-performance computing.

The manuscript is well-written, and the results are worth publication. Before recommending it for publication in Nature Communications, this referee would like to know the authors' response to the following questions/comments.

Q1. Do you have only +/- 10 mT data for self-switching of magnetization in PtCo? Do you have higher field data?

Reply: We thank the reviewer for this comment. We have added additional self-switching data in PtCo with the application of higher in-plane fields, as shown in **Figure R9**. We find no self-switching behavior in PtCo, up to $H_x = \pm 50$ mT. We have also included these results in revised Supplementary Figure 4.

Figure R9. Absence of current switching for PtCo single layer under various μ_0H_x .

Q2. How did you determine I_c ? What are the error bars?

Reply: The critical current I_c is a threshold current at which magnetization switching

occurs. In Hall bar devices (Fig. 2), I_c is defined as the current at which the normalized anomalous Hall resistance reaches 0. The definition of I_c is clarified in revised manuscript (located at page 6).

To check the reproducibility of I_c , we have also carried out the current switching measurements for multiple times. **Figure R10** shows the I_c for one representative device in ten successive measurements, revealing that the average variation of I_c is smaller than 2%. The error bar of I_c is determined by the standard deviation of these measurements. These error bars of I_c have been included in the manuscript. It's important to note that the modulation of I_c by voltage (~14%) is much larger than the variation of I_c . These results and discussions are also included in revised manuscript (located at page 6) and Supplementary Materials (see Supplementary Note 4 and Supplementary Figure 5).

Figure R10. Variation of threshold current I_c for magnon-torque-induced magnetization switching of the 11 nm SrRuO₃/120 nm BiFeO₃/PtCo samples in ten successive switching.

Q3. It is not clear to the reader why SrRuO₃/BiFeO₃/NiFe is used to determine magnon torque efficiency. Can you explain?

Reply: The magnon torque efficiency in this work was determined by the standard spin torque measurement techniques including ST-FMR and SHHV. These measurement techniques, especially for the ST-FMR, normally requires that the ferromagnetic layer has an in-plane magnetic anisotropy, which is the reason why we used the NiFe as the ferromagnetic overlayer to determine the magnon torque efficiency. The two methods provide consistent results as shown in our manuscript.

Moreover, since the PtCo is used for the demonstration of perpendicular magnetic switch, we have also conducted the additional SHHV measurements on

SrRuO₃/BiFeO₃/PtCo samples to determine the magnon torque efficiency with PtCo ferromagnetic overlayer, as shown in **Figure R11**. The measurement geometry is shown in **Figure R11a** below, as an ac current I_{AC} is applied along the x-direction, and an alternating magnon current is injected into PtCo layer, exerting a damping-like magnon torque effective field ($H_{m,DL}$) to induce the oscillation of PtCo magnetization. When the in-plane magnetic field H_x is larger than anisotropic field H_K , the magnetization vector M is aligned along the x-direction, leading to the SHH resistance to be written as [*Phys. Rev. Lett.* 123, 207205 (2019); *Nat. Commun.* 12, 4555 (2021)]:

$$R_{xy}^{2\omega} = \frac{R_{AHE}}{2} \frac{H_{m,DL}}{|H_x| - H_K} + R_{PHE} \frac{H_{FL}}{|H_x|} + R_{thermal}$$

Where R_{AHE} and R_{PHE} are the AHE and planar Hall resistances; $R_{thermal}$ is the thermal contribution from anomalous Nernst and spin Seebeck effects. Since $R_{PHE} \ll R_{AHE}$, the second term can be neglected. Thus, we can estimate the effective field H_{DL} via fitting the $R_{xy}^{2\omega}$ data in the large in-plane field regime.

We then apply I_{AC} along x direction with in-plane field parallel to the current in sample 11 nm SrRuO₃/120 nm BiFeO₃/PtCo tri-layer. The results for first and second harmonic hall resistance are shown in **Figure R11b** and **R11c** below. By changing the amplitude of current density (J) and fitting the second harmonic resistance, as shown in **Figure R11d** below, the $H_{m,DL}/J$ is estimated to be 0.077 mT per 10^6 A/cm². The damping-like spin-torque efficiency can be estimated as $\xi_{DL} = \frac{2e}{\hbar} M_s t_{Co} \frac{H_{DL}}{J_e}$. Here, we measure the saturated magnetization of Co as $M_s = 1100$ emu/cm³ consistent with reported values (*Phys. Rev. B* 99, 134421 (2019)), see **Figure R11e** below, and $t_{Co} = 1.6$ nm for the thickness of Co. The damping-like torque efficiency $\xi_{m,DL}$ is calculated to be around 0.041, which is within the same order of magnitude with the results measured by ST-FMR (0.012) and SHHV (0.027) in SrRuO₃/BiFeO₃/NiFe tri-layer. We have included the discussion and results in the revised manuscript (located on page 7) and Supplementary Materials (see Supplementary Note 8 and Supplementary Figure 8).

Figure R11. The magnon-torque efficiency in 11 nm SrRuO₃/120 nm BiFeO₃/PtCo heterostructures. **a**, The schematic of second harmonic measurements. The magnetization M is oscillated by $H_{m,DL}$. The blue arrow indicates the direction of magnetization M . The experimental results of $R_{xy}^{1\omega}$ and $R_{xy}^{2\omega}$ are shown in **b** and **c**, respectively. The blue circles are the experimental data, and the red lines are the fitting curves. **d**, the effective field $H_{m,DL}$ as a function of current density. The blue circles are experimental results, and red line represents the linear fitting. **e**, The out-of-plane magnetization hysteresis loop measurement of 11 nm SrRuO₃/120 nm BiFeO₃/PtCo heterostructure stack on DyScO₃. A paramagnetic background from DyScO₃ substrate has been removed.

Q4. Although the authors mention that efficiency is comparable to the previous studies they cited, the efficiency seems smaller than the literature. Is there any reason for the lower efficiency?

Reply: We thank the reviewer for this insightful comment. The reported spin torque efficiency in SrRuO₃-based heterostructures can vary significantly depending on various factors such as measurement techniques, ferromagnetic overlayers, sample quality, substrates-induced strain, octahedral distortion and artifacts from the dielectric property of substrate (*Phys. Rev. Appl.* 21, 024021 (2024)). To provide a comprehensive comparison, we have summarized the spin torque efficiency from our results and values reported in the literature in **Table 3** below. Our estimated magnon torque efficiency ranges from 0.012~0.027 for SrRuO₃-based heterostructures on DyScO₃ substrate, which is comparable to the previous reported SOT efficiency of SrRuO₃ on similar orthorhombic substrates (in Ref. *Adv. Mater.* 33, 2007114 (2021))

and *Adv. Func. Mater.* 31, 2100380 (2021)). This issue is addressed in our revised manuscript, see page 7.

Table 3. The summary of spin torque efficiency from our work and references

Heterostructure	Substrate	Spin torque efficiency	References
SrRuO ₃ /NiFe	SrTiO ₃ /KTaO ₃ (cubic)	0.15	Adv. Mater. 33, 2007114 (2021)
SrRuO ₃ /NiFe	SrTiO ₃ /KTaO ₃ (cubic)	0.62~1.07	Adv. Func. Mater. 31, 2100380 (2021)
SrRuO ₃ /NiFe	NdGdO ₃ (orthorhombic)	0.02~0.04	Adv. Mater. 33, 2007114 (2021)
SrRuO ₃ /NiFe	DyScO ₃ (orthorhombic)	0.04	Adv. Func. Mater. 31, 2100380 (2021)
SrRuO₃/BiFeO₃/NiFe	DyScO₃ (orthorhombic)	0.012~0.027	This work

In-situ Voltage control of magnon transport

Q5. The value of I_c you determined in Figure 2e is larger than the minimum I_w of 8 mA in Figure 3d. Can you explain the difference?

Reply: We appreciate the reviewer's observation and would like to clarify the difference between the threshold currents (I_c) observed in Figures 2e and 3d. In our experiments, devices for magnon-torque induced current switching measurements (shown in Fig. 2) were patterned into Hall bars, in which the charge current flows along both NiFe and SrRuO₃ layer. On the other hand, in devices for voltage-controlled magnon torque (shown in Fig. 3), the charge current only flows along the SrRuO₃ layer. Thus, the I_w for these two devices is different.

In Figure 2, we show the current switching of PtCo in a Hall bar device, in which the current density is calculated according to parallel resistance model. Our control samples provide the resistivity for the calculation: the resistivity of 8.3 nm PtCo layer on Si/SiO₂ substrate ρ_{PtCo} is estimated to be about 120 $\mu\Omega\cdot\text{cm}$, and the resistivity of 11 nm SrRuO₃ layer on DyScO₃ substrate ρ_{SRO} is about 325 $\mu\Omega\cdot\text{cm}$. The current density in SrRuO₃ layer is determined by the following expression:

$$j_{SRO} = \frac{I_w}{W} \times \frac{\rho_{PtCo}}{\rho_{SRO} \cdot d_{PtCo} + \rho_{PtCo} \cdot d_{SRO}}$$

where the I_w is the magnitude of current pulse, $W = 16 \mu\text{m}$ is the width of device, $t_{\text{PtCo}} = 8.3 \text{ nm}$ is the thickness of PtCo layer, and $d_{\text{SRO}} = 11 \text{ nm}$ is the thickness of SrRuO₃ layer. In Figure 2, the threshold current $I_c = 16 \text{ mA}$, corresponding to $j_{\text{SRO}} \approx 3.0 \times 10^6 \text{ A/cm}^2$.

In Figure 3 we show the current switching of PtCo in pillar devices. The current density is calculated as $j_{\text{SRO}} = \frac{I_w}{W \cdot d_{\text{SRO}}} \approx 2.4 \times 10^6 \text{ A/cm}^2$ ($I_c = 8 \text{ mA}$, $W = 30 \mu\text{m}$) because there is no current shunting in PtCo layer in these devices. We find the critical current densities in the Hall bar (Figure 2) and pillar devices (Figure 3) are comparable. We also include these discussions and calculations in our revised manuscript (located on page 6) and Supplementary Materials (see Supplementary Note 5).

Q6. Why the colors are different in Fig 3e (lower panel)?

Reply: We appreciate the reviewer's observation regarding the colors in MOKE images. In Fig. 3e, we show the raw data of MOKE microscopy, in which the comparative color contrasts indicate the out-of-plane component of magnetization M_z . It's important to note that the color contrasts in MOKE microscopy are influenced by background subtraction, which can result in varying grayscale levels across different images. This is common in MOKE microscope measurements. The key aspect to focus on in Figure 3e is the change in color contrast within the device pillars before and after the application of current pulses. These changes indicate magnetization switching events, which are the primary focus of our investigation. We have included this clarification in our revised manuscript located at page 8 and page 10.

Q7. Can the difference in I_c with the direction of P_z depends on the domain distribution of BiFeO₃?

Reply: We agree with the reviewer's comment, which is actually the proposed mechanism for the modulation of I_c in the two-variant BiFeO₃ as shown in Supplementary Fig. 2. According to our Supplementary Fig. 2, the PFM results clearly show the different domain distribution of BiFeO₃ before and after ferroelectric polarization switching, similar to previous reports. In addition, in the two types domain, the spin cycloid direction is different, leading to modulation of magnon transport and giving rise to the modulation of I_c , see Supplementary Figure 1.

Q8. Have you tried the same with a single-domain sample?

Reply: We thank the reviewer for this insightful comment. We have replicated the magnetization switching experiments in mono-domain BiFeO₃ sample with a the structure of SrRuO₃/BiFeO₃/PtCo on SrTiO₃ substrate, see Supplementary Fig. 12. We observed a clear modulation of I_c, albeit with a smaller ratio compared to the two-variant-domain BiFeO₃ sample. This discrepancy can be understood as a different domain structure and spin cycloid configuration in BiFeO₃ due to the confinements of strain and miscut of substrate. The mono-domain SrTiO₃/SrRuO₃/BiFeO₃/PtCo sample presents a singular spin cycloid direction. Upon ferroelectric switching, the cycloid plane can be modulated as previously demonstrated (*Nat. Commun.* 8, 1583 (2017).). The relatively modest modulation ratio of I_c in the mono-domain sample likely stems from the change in the cycloid plane following ferroelectric switching. These findings underscore the potential for achieving a higher modulation ratio through deliberate engineering of the antiferromagnetic structure, an avenue that warrants further investigation. This issue is also included in revised manuscript (located at page 13) and Supplementary Materials (see Supplementary Note 13 and Supplementary Figure 14).

Reconfigurable Boolean logic operations

Q9. You did not define I_{c1} and I_{c2} before discussing them on page 10. What are their values for the current setup? Will those change from sample to sample? If so, which sample parameters will affect I_{c1} and I_{c2}?

Reply: We thank the reviewer for this comment. The I_{c1} (I_{c2}) is defined as threshold switching current with upward (downward) ferroelectric polarization. We have added this definition to the manuscript in page 9 for clarity. For devices with two-variant domain structure, the I_{c1} and I_{c2} is about 7.0 and 8.0 mA, respectively, as shown in Fig. 3. For devices with mono-domain structure, the I_{c1} and I_{c2} is about 12 and 12.5 mA, respectively (Supplementary Fig. 14). This is clarified in our revised manuscript, see page 11 and Supplementary Materials. We also conduct voltage control magnon torque switching experiments for multiple devices, finding modulation ability about 10%~14% in two-variant sample and 2%~4% for mono-domain sample, respectively. We speculate that the ferroelectric and antiferromagnetic domain structure would mainly affect I_{c1} and I_{c2}. This discussion is included in revised manuscript (located at page 13) and Supplementary Materials (see Supplementary Note 13).

Q10. Does the Domain structure variation affect I_{c1} and I_{c2}? If so, does the repetitive measurements affect I_{c1} and I_{c2}?

Reply: We thank the reviewer for this insightful comment. We have observed that the modulation ratio differs between device with a two-variant domain BiFeO₃ (Fig. 3) and those with a mono-domain BiFeO₃ (Supplementary Fig. 14), suggesting that the

domain structure variation does indeed affect I_{c1} and I_{c2} . In addition, as shown in Supplementary Fig. 14, we have carried out measurements on BiFeO₃ devices by repetitively switching the ferroelectric polarization. We found that the modulation of I_{c1} and I_{c2} is reversible. This discussion is also included in revised manuscript (see page 13) and Supplementary Materials (see Supplementary Note 13).

Q11. Do the Authors have an estimate of the Mz?

Reply: We thank the reviewer for this insightful comment. We conduct the out-of-plane magnetization hysteresis loop measurement of 11 nm SrRuO₃/120 nm BiFeO₃/PtCo heterostructure using a vibrating sample magnetometer. The results are shown in **Figure R12** with the paramagnetic background from DyScO₃ substrate removed. We find the saturated magnetization of Co is about 1100 emu/cm³, consistent with the reported value (*Phys. Rev. B* 99, 134421 (2019)). This information has been addressed in revised manuscript (located at page 4) and included in Supplementary Materials (see Supplementary Figure 3).

Figure R12. The out-of-plane magnetization hysteresis loop measurement of a 11 nm SrRuO₃/120 nm BiFeO₃/PtCo heterostructure stack on DyScO₃ substrate. Paramagnetic background from DyScO₃ substrate has been removed.

REVIEWERS' COMMENTS

Reviewer #1 (Remarks to the Author):

I think that the authors will address all comments raised by the reviewers as much as possible at present. I recommend this paper be suitable for publication in Nature Communications in this revised form.

Reviewer #2 (Remarks to the Author):

The authors have responded to my comments properly, and my concerns have been resolved. Therefore, I would like to recommend this work for publication.

Reviewer #3 (Remarks to the Author):

The authors have addressed my concerns in the response and revised manuscript. I recommend it for publication.

Reviewer #4 (Remarks to the Author):

The authors responded to my referee's comments comprehensively and updated the manuscript appropriately. They also provided a detailed rebuttal for my comments. This referee is satisfied with the rebuttal and changes made in the manuscript. Additionally, the authors replied and made detailed changes to comments from other referees. The manuscript is well-written and comprehensive, and the results are worth publication. Therefore, I recommend the manuscript for publication in Nature Communications.

Response to Reviewers

REVIEWER 1 COMMENTS:

I think that the authors will address all comments raised by the reviewers as much as possible at present. I recommend this paper be suitable for publication in Nature Communications in this revised form.

Reply: We thank this referee for now in the position of supporting the publication of our manuscript in Nature Communications.

REVIEWER 2 COMMENTS:

The authors have responded to my comments properly, and my concerns have been resolved. Therefore, I would like to recommend this work for publication.

Reply: We thank this referee for now in the position of supporting the publication of our manuscript in Nature Communications.

REVIEWER 3 COMMENTS:

The authors have addressed my concerns in the response and revised manuscript. I recommend it for publication.

Reply: We thank this referee for now in the position of supporting the publication of our manuscript in Nature Communications.

REVIEWER 4 COMMENTS:

The authors responded to my referee's comments comprehensively and updated the manuscript appropriately. They also provided a detailed rebuttal for my comments. This referee is satisfied with the rebuttal and changes made in the manuscript. Additionally, the authors replied and made detailed changes to comments from other referees. The manuscript is well-written and comprehensive, and the results are worth publication. Therefore, I recommend the manuscript for publication in Nature Communications.

Reply: We thank this referee for now in the position of supporting the publication of our manuscript in Nature Communications.